Comparison of nutritional supplements in improving glycolipid metabolism and endocrine function in polycystic ovary syndrome: a systematic review and network meta-analysis

Hu Xinyin 1
Wang Wanyi 1
Su Xuhan 1
Peng Haoye 1
Tan Zuolin 1
Li Yunqing 2
Huang Yuhua 3 hyuhua@163.com
1 Beijing University of Chinese Medicine , Beijing , China
2 Capital Medical University , Beijing , China
3 Beijing Hospital of Traditional Chinese Medicine, Capital Medical University , Beijing , China
Dong Peixin
Electronic publication date: 2023 Nov 13
Publication date: 2023
Volume: 11
Electronic Location ID: e16410
Received 2023 Aug 21; Accepted 2023 Oct 15
Copyright: © 2023 Hu et al.
Copyright year: 2023
Copyright holder: Hu et al.
License: This is an open access article distributed under the terms of the Creative Commons Attribution License, which permits unrestricted use, distribution, reproduction and adaptation in any medium and for any purpose provided that it is properly attributed. For attribution, the original author(s), title, publication source (PeerJ) and either DOI or URL of the article must be cited.
License URL: https://creativecommons.org/licenses/by/4.0/

Keywords: Nutritional supplements, Polycystic ovary syndrome, Network meta-analysis, Endocrine function, Glycolipid metabolism

Funding: This study was supported by the Beijing Traditional Chinese Medicine Science and Technology Development Fund project (JJ-2020-47). The funders had no role in study design, data collection and analysis, decision to publish, or preparation of the manuscript.

==============================
Objective

To explore the comparative effectiveness of nutritional supplements in improving glycolipid metabolism and endocrine function in patients with polycystic ovary syndrome (PCOS).

Method

Randomized controlled clinical trials on the effects of nutritional supplements in PCOS patients were searched in PubMed, Embase, Cochrane Library, and Web of Science from their establishments to March 15, 2023. Then, literature screening, data extraction, and network meta-analysis were performed. This study was registered at PROSPERO (registration number CRD 42023441257).

Result

Forty-one articles involving 2,362 patients were included in this study. The network meta-analysis showed that carnitine, inositol, and probiotics reduced body weight and body mass index (BMI) compared to placebo, and carnitine outperformed the other supplements (SUCRAs: 96.04%, 97.73%, respectively). Omega-3 lowered fasting blood glucose (FBG) (SUCRAs: 93.53%), and chromium reduced fasting insulin (FINS) (SUCRAs: 72.90%); both were superior to placebo in improving insulin resistance index (HOMA-IR), and chromium was more effective than Omega-3 (SUCRAs: 79.99%). Selenium was potent in raising the quantitative insulin sensitivity index (QUICKI) (SUCRAs: 87.92%). Coenzyme Q10 was the most effective in reducing triglycerides (TG), total cholesterol (TC), and low-density lipoprotein cholesterol (LDL-C) levels (SUCRAs: 87.71%, 98.78%, and 98.70%, respectively). Chromium and probiotics decreased TG levels, while chromium and vitamin D decreased TC levels. No significant differences were observed in high-density lipoprotein cholesterol (HDL-C), total testosterone (TT), sex-hormone binding globulin (SHBG), and C-reactive protein (CRP) between nutritional supplements and placebo.

Conclusion

Carnitine was relatively effective in reducing body mass, while chromium, Omega-3, and selenium were beneficial for improving glucose metabolism. Meanwhile, coenzyme Q10 was more efficacious for improving lipid metabolism. However, publication bias may exist, and more high-quality clinical randomized controlled trials are needed.

Introduction

Polycystic ovary syndrome (PCOS) is a common gynecologic endocrine disorder, which affects approximately 8 to 15% of women of childbearing age (Hoeger, Dokras & Piltonen, 2021). This disorder is heterogeneous and characterized by androgen excess (hirsutism and/or hyperandrogenism) and ovarian dysfunction (ovulation disorders and/or polycystic ovary morphology) (Escobar-Morreale, 2018). The pathogenesis of PCOS is not fully understood, but it may be associated with hyperandrogenism, insulin resistance, obesity, metabolic abnormalities, and inflammation (Li et al., 2019). Patients with PCOS are at a higher risk of developing Type 2/gestational diabetes mellitus, atherosclerotic dyslipidemia, systemic inflammation, nonalcoholic fatty liver, hypertension, and coagulation disorders, significantly affecting their health and quality of life (Anagnostis, Tarlatzis & Kauffman, 2018).

Given the complex pathogenesis of PCOS, certain nutritional supplements are increasingly used for the management of PCOS due to their potential benefits in improving endocrine and metabolic disorders, in addition to conventional pharmacological treatments. Nutritional supplements have been demonstrated to alleviate oxidative stress in PCOS, thereby reducing the risks of disease progression and cardiovascular events. Furthermore, dietary supplements such as vitamins, minerals, and probiotic supplements effectively relieve PCOS-related symptoms (Dubey et al., 2021). Supplementation with natural molecules such as inositol, vitamin E, vitamin D, and omega-3 may help relieve pathological signs of PCOS, like immature oocytes, insulin resistance, hyperandrogenism, oxidative stress, and inflammation (Iervolino et al., 2021).

Several recent meta-analyses have examined the effects of nutritional supplements such as vitamins and minerals in patients with PCOS. A study conducted by Zhang et al. (2021) showed that inositol supplementation can raise SHBG and improve glycolipid metabolism; vitamin E may lower TT and raise SHBG; coenzyme Q10 alone or in combination with vitamin E can help reduce HOMA-IR. A study by Zhao et al. (2023) showed that selenium can improve total antioxidant capacity (TAC) in patients with PCOS, and selenium supplementation may ameliorate follicular quality of patients with PCOS who have poor follicles due to oxidative stress or who wish to undergo in-vitro fertilization. A study conducted by Gong et al. (2023) on seven randomized controlled studies showed that carnitine improves ovulation, clinical pregnancy rates, insulin resistance, and BMI in women with PCOS. A study conducted by Fazelian et al. (2017) suggested that chromium supplementation may be beneficial for reducing BMI, fasting insulin, and free testosterone in women with PCOS.

However, direct comparisons of different nutritional supplements for PCOS are lacking. The effects of nutritional supplements in PCOS patients need to be comprehensively analyzed. Therefore, this systematic review and network meta-analysis, based on the results of relevant randomized controlled trials, aimed to compare the effects of various nutritional supplements, including selenium, chromium, carnitine, inositol, coenzyme Q10, Omega-3, probiotics, vitamin D, and vitamin E in improving the endocrine function and glycolipid metabolism of individuals with PCOS.

Materials and Methods

Search strategy

This study has been registered at the International Prospective Register of Systematic Reviews (PROSPERO) (https://www.crd.york.ac.uk/PROSPERO/, registration number CRD 42023441257). PubMed, Embase, Cochrane Library, and Web of Science were searched from their establishments to March 15, 2023, and the language of literature was limited to English. To accurately and comprehensively gather relevant literature, subject terms and free words were combined to design specific keywords. The keywords used for retrieval mainly included disease name, nutritional supplement ingredients and study type, such as “Polycystic Ovary Syndrome”, “Inositol”, “coenzyme Q10”, “Vitamin E”, “omega-3”, “Vitamin D”, “Chromium”, “Selenium”, “Carnitine”, “Probiotics” and “randomized controlled trial”. The specific search strategy is shown in Table S1. The reference lists of published systematic reviews were also manually searched to avoid missing any potentially eligible studies. The literature search was conducted by two researchers, ZT and YL. Disagreements were resolved through discussion or consultation with XH.

Inclusion and exclusion criteria

The inclusion criteria were as follows: (1) Population: female patients aged 18–49 years diagnosed with PCOS according to the Rotterdam Criteria (The Rotterdam ESHRE/ASRM-Sponsored PCOS Consensus Workshop Group, 2003) or the National Institutes of Health Criteria (Zawadski & Dunaif, 1992). (2) Interventions: Selenium, chromium, carnitine, inositol, coenzyme Q10, Omega-3, probiotics, vitamin D, or vitamin E was used as in the experimental group, and placebo or one of the above nutritional supplements was used in the control group for eight weeks at least. No other medications were used in the intervention and control groups. (3) Study type: Randomized controlled clinical trial. (4) Outcome indicators: Primary outcomes consisted of body mass and markers of glucose and lipid metabolism, including body weight, BMI, FBG, FINS, HOMA-IR, QUICKI, TG, and TC. Secondary outcomes included LDL-C, HDL-C, TT, SHBG, and CRP. The following studies were excluded: (1) animal or cell experiments, case reports, scientific experiment plans, reviews, letters, editorials, and conference papers. (2) Literature with missing or grossly erroneous study data. (3) Only one copy of the literature should be retained.

Literature selection and data extraction

The searched papers were imported into EndNote 20.5. The title and abstracts were independently screened first by two investigators, XH and WW. Then the full texts were read for a second screening. Disagreements were resolved through discussion or consultation with a third investigator, YH. Two investigators used Excel 2016 to independently extract data from the final included studies. The extracted data included first author, year of publication, country, randomization and blinding, intervention and control, duration of the treatment, profile of the study population, and outcome indicators.

Quality assessment

The Cochrane Risk of Bias Assessment Tool (RoB2.0) (Higgins et al., 2022) was used to assess the risk of bias in included studies in six domains: poor randomization process, deviation from the defined interventions, missing outcome data, outcome measurement, selective reporting of outcomes, and other sources. Two reviewers, XH and WW, independently rated each domain as having a low risk, high risk, or some concerns. Disagreements were resolved through discussion or consultation with a third investigator, YH. The quality assessment results were presented in a risk-of-bias plot.

Statistical analysis

The continuous variables were presented as weighted mean differences (MD) with 95% CIs. We firstly fitted the Bayesian hierarchical random-effects model for the multiple comparisons of different treatment plans for PCOS, considering the heterogeneity among the trials (Dias et al., 2013; Mills, Thorlund & Ioannidis, 2013). We used the R 4.2.1 (R Core Team, 2022) software and Stata 15.1 (https://www.stata.com/stata15/) software to perform all calculations and obtain all charts. With R 4.2.1 software, Markov chain Monte Carlo (MCMC) simulation was conducted using Bayesian inference, which was based on the theory of likelihood function and some prior assumptions. Then, we set 500,000 in iterations and 20,000 in annealing in order to investigating the posterior distributions of the interrogated nodes (Bois, 2013; Dias et al., 2012; Hamra, MacLehose & Richardson, 2013). The node splitting method was used to evaluate local inconsistency for outcomes with closed loops. The relationships between different treatments were presented in the form of a network diagram. At the same time, we used a comparison-adjusted funnel plot to detect potential publication bias (Chaimani et al., 2013; Whegang Youdom, Tahar & Basco, 2017). In addition, we ranked the examined treatments by utilizing Surface Under Cumulative Ranking Probabilities (SUCRA) values, with SUCRA values ranging from 0 to 1. A higher SUCRA value denotes a higher ranking (Rücker & Schwarzer, 2015; Trinquart et al., 2016). We generated a league table to display the comparisons between each pair of interventions in each outcome.

Results

Search results and study characteristics

The initial search yielded 965 articles. A total of 439 duplicates were removed. After a preliminary reading of titles and abstracts, 442 studies were excluded. The full texts of the remaining studies were downloaded and read to exclude ineligible studies. Finally, 41 articles were included. The detailed screening process is shown in Fig. 1.

Figure 1 Literature screening process.

The 41 included studies (Ahmadi et al., 2017; Al-Bayyari et al., 2021; Amini et al., 2020; Arab et al., 2022; Ardabili, Gargari & Farzadi, 2012; Ashoush et al., 2016; Bader, Althanoon & Raoof, 2022; Bonakdaran et al., 2012; Dastorani et al., 2018; Donà et al., 2012; Esmaeilinezhad et al., 2019; Esmaeilinezhad et al., 2020; Heidar et al., 2020; Hosseinzadeh et al., 2016; Irani et al., 2015; Jafari-Sfidvajani et al., 2018; Jamilian et al., 2017; Jamilian & Asemi, 2015; Jamilian et al., 2018; Jamilian et al., 2015; Javed et al., 2019; Karamali et al., 2018; Karamali & Gholizadeh, 2022; Karimi et al., 2020; Karimi et al., 2018; Kaur et al., 2022; Lucidi et al., 2005; Maktabi, Chamani & Asemi, 2017; Modarres, Asemi & Heidar, 2022; Modarres et al., 2018; Mohammadi et al., 2012; Nadjarzadeh et al., 2015; Rafraf et al., 2012; Rahimi-Ardabili, Pourghassem Gargari & Farzadi, 2013; Rahmani et al., 2018; Raja-Khan et al., 2014; Rashidi et al., 2020; Seyyed Abootorabi et al., 2018; Shoaei et al., 2015; Siavashani et al., 2018; Trummer et al., 2019) were conducted in nine countries. There were a total of 2,362 patients, with 1,214 in the intervention group and 1,148 in the control group. The intervention group used selenium supplements in six studies, chromium supplements in five studies, carnitine in one study, inositol in one study, coenzyme Q10 in three studies, Omega-3 in four studies, probiotics in nine studies, and vitamin D in 12 studies. The control group was given a placebo in all the studies. The basic characteristics of the included literature are presented in Table S2.

Assessment of the risk of bias

The risk-of-bias assessment results of the 41 included studies are shown in Fig. 2. The risk of bias in the randomization process was assessed as “some concerns” for four studies due to a lack of randomized assignment or allocation concealment, and low for the other 37 studies. The risk of bias in deviation from the defined interventions was assessed as “having some concerns” for two studies due to no description of blinding, and low for the other 39 studies. All studies were at low risk of bias in missing outcome data and outcome measurement. The risk of bias in selective reporting was unclear and assessed as “some concerns” for all studies. The risk of other sources of bias was low for all included studies as no such risk was found. Overall, the risk of bias in the included literature was low.

Figure 2 Assessment figure of risk of bias.

Network diagrams

The 41 included studies covered eight nutritional supplements: selenium, chromium, carnitine, inositol, coenzyme Q10, Omega-3, probiotics, and vitamin D. The network for direct comparison of nutritional supplements and placebo is shown in Fig. 3. In the figure, the thickness of the lines is proportional to the number of studies included in the pairwise comparison, and the diameter of the circles is proportional to the number of participants who received the intervention.

Figure 3 Network structure diagram.

Outcomes of the network meta-analysis

The cumulative probability values of the impact of different interventions on outcome indicators are shown in Table 1, and the cumulative probability line charts are shown in Fig. 4.

Table 1 Cumulative probability values.

T	Weight	BMI	FBG	FINS	HOMA-IR	QUICKI	TG	TC	LDL-C	HDL-C	TT	SHBG	CRP	
Carnitine	0.96	0.98	0.00	0.00	0.00	0.00	0.00	0.00	0.00	0.00	0.00	0.00	0.00	
Chromium	0.27	0.36	0.45	0.73	0.80	0.68	0.87	0.68	0.61	0.48	0.05	0.00	0.72	
CoQ10	0.19	0.24	0.55	0.00	0.00	0.00	0.88	0.99	0.99	0.33	0.60	0.63	0.54	
Inositol	0.91	0.89	0.36	0.81	0.71	0.00	0.00	0.00	0.00	0.00	0.00	0.00	0.00	
Omega3	0.42	0.54	0.94	0.63	0.72	0.51	0.23	0.55	0.54	0.72	0.76	0.41	0.64	
Placebo	0.22	0.23	0.27	0.14	0.12	0.11	0.15	0.16	0.16	0.39	0.37	0.31	0.39	
Probiotic	0.69	0.70	0.70	0.59	0.57	0.36	0.67	0.28	0.56	0.86	0.72	0.86	0.16	
Selenium	0.40	0.31	0.30	0.18	0.30	0.88	0.32	0.21	0.20	0.41	0.32	0.28	0.00	
Vitamin D	0.44	0.26	0.44	0.42	0.29	0.47	0.38	0.63	0.45	0.31	0.68	0.51	0.55	
Note:

Bold values represent the SUCRAs of intervention measures ranking in the top 3 for each outcome indicator.

Figure 4 Cumulative probability line chart.

Body mass

Twenty-six studies reported body mass. The results showed that carnitine (MD = 2.4, 95% CI [1.7–3.11]), inositol (MD = 2.08, 95% CI [0.99–3.12]), and probiotics (MD = −0.6, 95% CI [−1.09 to −0.11]) were superior to placebo in reducing weight, with statistically significant differences. Based on the cumulative probability results, carnitine (SUCRAs: 96.04%), inositol (SUCRAs: 91.06%), and probiotics (SUCRAs: 68.69%) were likely to be the most effective in lowering body mass.

Twenty-nine studies reported BMI. The results showed that carnitine (MD = −0.99, 95% CI [−1.26 to −0.74]), inositol (MD = −0.78, 95% CI [−1.16 to −0.39]), and probiotics (MD = 0.25, 95% CI [0.07–0.46]) outperformed placebo in reducing BMI, with statistically significant differences. Based on the cumulative probability results, carnitine (SUCRAs: 97.73%), inositol (SUCRAs: 89.34%), and probiotics (SUCRAs: 69.70%) were likely to be the most effective in lowering BMI.

Glucose metabolism

Twenty-five studies reported FBG. The results showed that Omega-3 (MD = 6.71, 95% CI [1.89–11.81]) was superior to placebo in lowering fasting blood glucose, with statistically significant differences. Based on the cumulative probability results, Omega-3 (SUCRAs: 93.53%), probiotics (SUCRAs: 69.55%), and coenzyme Q10 (SUCRAs: 54.86%) were likely to be the most effective for lowering fasting blood glucose.

Twenty-three studies reported FINS. The results showed that chromium (MD = −2.54, 95% CI [−5.17 to −0.21]) was more effective than placebo in reducing fasting insulin levels, with statistically significant differences. Based on the cumulative probability results, Inositol (SUCRAs: 80.71%), Chromium (SUCRAs: 72.90%), and Omega-3 (SUCRAs: 62.88%) were possibly the three most effective in lowering fasting insulin.

Twenty-three studies reported HOMA-IR. The results showed that chromium (MD = 0.91, 95% CI [0.07 to 1.83]) and Omega-3 (MD = 0.75, 95% CI [0.04 to 1.45]) improved HOMA-IR compared to placebo, with statistically significant differences. Based on the cumulative probability results, chromium (SUCRAs: 79.99%), Omega-3 (SUCRAs: 71.69%), and inositol (SUCRAs: 70.86%) may be the most effective in improving HOMA-IR.

Fifteen studies reported QUICKI. The results showed that selenium (MD = −0.02, 95% CI [−0.04 to 0]) was more beneficial than placebo in improving QUICKI, with statistically significant differences. Based on the cumulative probability results, selenium (SUCRAs: 87.92%), chromium (SUCRAs: 67.53%), and Omega-3 (SUCRAs: 51.38%) were likely to be the most effective in improving QUICKI.

Lipid metabolism

Eighteen studies reported on TG. The results showed that chromium (MD = 23.76, 95% CI [8.33–38.89]), coenzyme Q10 (MD = 25.59, 95% CI [2.82–48.31]), and probiotics (MD = −13.22, 95% CI [−25.24 to −2.06]) outperformed placebo in lowering TG levels, with statistically significant differences. Based on the cumulative probability results, coenzyme Q10 (SUCRAs: 87.71%), chromium (SUCRAs: 87.50%) and probiotics (SUCRAs: 66.85%) were likely to be the most effective nutritional supplements for reducing TG levels.

Eighteen studies reported TC. The results showed that chromium (MD = −11.13, 95% CI [−22.13 to 0]), coenzyme Q10 (MD = −31.74, 95% CI [−49.52 to −13.78]), and vitamin D (MD = 9.76, 95% CI [1.28–16.12]) were superior to placebo in reducing TC levels, with statistically significant differences. Based on the cumulative probability results, coenzyme Q10 (SUCRAs: 98.78%), chromium (SUCRAs: 67.64%), and vitamin D (SUCRAs: 63.31%) were likely to be the most effective in lowering TC levels.

Seventeen studies reported on LDL-C. The results showed that coenzyme Q10 (MD = 30.11, 95% CI [11.38–49.12]) was more effective than placebo in lowering LDL-C levels, with statistically significant differences. Based on the cumulative probability results, coenzyme Q10 (SUCRAs: 98.70%), chromium (SUCRAs: 60.77%), and probiotics (SUCRAs: 55.96%) were likely to be the most effective in reducing LDL-C levels.

Sixteen studies reported on HDL-C. The results showed no significant differences in HDL-C between supplements and placebo. Based on the cumulative probability results, probiotics (SUCRAs: 85.60%), Omega-3 (SUCRAs: 71.01%), and chromium (SUCRAs: 48.04%) were likely to be the most effective in raising HDL-C levels.

Hyperandrogenemia

Sixteen studies reported TT while ten studies reported SHBG. The results showed no statistical difference between all nutritional supplements and placebo in improving hyperandrogenism. Based on the cumulative probability results, Omega-3 (SUCRAs: 75.93%), probiotics (SUCRAs: 72.23%), and chromium (SUCRAs: 68.05%) were likely to have the best effects in reducing TT levels. Probiotics (SUCRAs: 85.86%), coenzyme Q10 (SUCRAs: 62.69%), and vitamin D (SUCRAs: 50.84%) were likely to have the best effects in improving SHBG levels.

Inflammatory response

Twelve studies reported CRP. The results showed no significant difference between all nutritional supplements and placebo in improving the inflammatory response. Based on the cumulative probability results, chromium (SUCRAs: 72.43%), Omega-3 (SUCRAs: 63.91%), and vitamin D (SUCRAs: 55.29%) were likely to have the best effect in lowering CRP levels.

Publication bias

Funnel plots were used to evaluate the publication bias of all outcome indicators. As shown in Fig. 5, the funnel plots of BMI, glucose, insulin, HOMA-IR, HDL-C, total testosterone, and CRP might be asymmetric with skewed distribution, suggesting that there may be a certain degree of publication bias.

Figure 5 Funnel plots.

Safety analysis

A total of 22 studies reported adverse reactions. Among these studies, 20 reported no significant adverse events, and two reported the symptoms and number of adverse reactions, as shown in Table S3. Overall, nutritional supplements may be highly safe with a low incidence of adverse reactions.

Discussion

This meta-analysis included 41 clinical randomized controlled studies with 2,362 patients to investigate the effects of various nutritional supplements in improving endocrine function and glucolipid metabolism in patients with PCOS. There were eight nutritional supplements, including selenium, chromium, carnitine, inositol, coenzyme Q10, Omega-3, probiotics, and vitamin D. According to the results, carnitine, inositol and probiotics reduced weight and BMI compared to placebo, and carnitine outperformed the other supplements. Omega-3 lowered fasting blood glucose, while chromium lowered fasting insulin; both were superior to placebo in improving HOMA-IR, and chromium was more effective than Omega-3. Selenium was effective in increasing QUICKI. Coenzyme Q10 was the most effective in lowering TG, TC, and LDL-C levels; chromium and probiotics lowered TG levels; chromium and vitamin D lowered TC levels. No significant differences were noted between nutritional supplements and placebo in HDL-C, TT, SHBG, and CRP.

Carnitine was more effective than other nutritional supplements in reducing weight and BMI. This finding seems to be in line with the findings of Gong et al. (2023). Carnitine is a quaternary amine in almost all animal species, which is usually introduced through food or synthesized at the cellular level. Carnitine has two enantiomeric forms: L-carnitine and D-carnitine. The former plays an essential role in cellular energy production, such as fatty acid uptake, β-oxidation, and regulation of glucose metabolism. It has been shown that serum total L-carnitine levels are significantly lower in non-obese PCOS women compared to healthy women. The decreased carnitine levels in circulation and tissue may be possibly due to impaired mitochondrial function, which is thought to be associated with the pathogenesis of insulin resistance (Petrillo et al., 2021). Oral administration of acyl L-carnitine attenuates ovarian dysfunction, while acyl L-carnitine combined with propionyl L-carnitine provides better activity. The molecular mechanisms of these effects include antioxidant/glycolytic activity and mitochondrial augmentation (Di Emidio et al., 2020). A study conducted by Talenezhad et al. (2020) suggested a nonlinear dose-response relationship between L-carnitine supplementation and weight reduction (P < 0.001). This indicates that 2,000 mg of L-carnitine daily was more effective for adults with PCOS.

In terms of glucose metabolism, chromium may reduce fasting insulin levels and insulin resistance, which is in line with the study of Fazelian et al. (2017). Moreover, a systematic review conducted by Heshmati et al. (2018) has indicated that chromium has a similar effect on insulin resistance reduction as metformin. In a study conducted by Chen et al. (2017) in PCOS model mice, serum and muscle Cr levels were negatively correlated with serum insulin concentrations, indicating that reduced serum and muscle Cr levels with hyperinsulinemia may be observed in PCOS patients. Some experimental studies (Chen et al., 2009) have revealed that chromium enhances skeletal muscle cell insulin signaling by participating in immunosuppression. Chromium supplementation can activate post-receptor insulin signaling, such as increasing the expression of insulin receptor substrate 1 (IRS1) and glucose transporter 4 (Glut 4), stimulating phosphatidylinositol 3-kinase (PI3-k) and protein kinase B (Akt) activity, downregulating c-Jun N-terminal kinase (JNK) activity, and decreasing IRS1 ubiquitylation and insulin resistance-associated IRS1 phosphorylation, thereby improving insulin resistance.

Omega-3 may be effective in lowering fasting blood glucose. Omega-3 is a type of polyunsaturated fatty acid, of which eicosapentaenoic acid (EPA) and docosahexaenoic acid (DHA) are the most biologically active. Omega-3 fatty acids have antioxidant, anti-inflammatory, anti-obesity, and insulin-sensitive properties. They can improve insulin sensitivity by decreasing the production of inflammatory cytokines, including tumor necrosis factor-α (TNF-α) and interleukin-6 (IL-6), as well as increasing the secretion of anti-inflammatory lipocalins (Iervolino et al., 2021). Omega-3 mediates insulin secretion from pancreatic β-cells by increasing the release of GPRs-mediated glucagon-like peptide 1 (GLP-1) from enteroendocrine L-cells, up-regulating the glucagon pathway and down-regulating other control pathways, which may play an important role in lowering fasting blood glucose (Bhaswant, Poudyal & Brown, 2015).

Selenium probably ranks first in terms of improving QUICKI. It is an essential trace element with significant antioxidant and anti-inflammatory functions. It inhibits the production of reactive oxygen species (ROS) by increasing the activity of selenoproteins. Selenium is also involved in metabolic functions. Decreased selenium concentrations in plasma in females with PCOS can lead to free radical accumulation and hyperandrogenism (Alesi et al., 2022). An animal experiment by Rabah et al. (2023) showed that in a polycystic ovary syndrome-insulin resistance (PCOS-IR) rat model, selenium nanoparticles (SeNPs) can enhance insulin sensitivity by modulating the phosphatidylinositol 3-kinase/protein kinase B (PI3K/Akt) pathway and activating kinases such as Akt in the insulin signaling cascade to trigger insulin-like effects. However, another study (Hosseinzadeh et al., 2016) reported a significant increase in insulin resistance after 12 weeks of 200 μg selenium supplementation per day in patients with PCOS. Therefore, selenium supplements should be cautiously used for patients with PCOS, and more large-scale studies are needed. In the original studies included in our analysis, selenium is administered at 200 μg per day for 8 weeks, which can provide some references.

As for improving lipid metabolism, coenzyme Q10 might have a more prominent effect in lowering the levels of TG, TC, and LDL-C, which is consistent with the findings of Zhang et al. (2023) CoQ10 is a lipophilic benzoquinone enriched in mammalian organs such as the heart, liver, and kidneys and it can be found in the membranes of virtually all mammalian cells (Zhang et al., 2018). CoQ10 has many vital cellular functions. It serves as an electron carrier in the mitochondrial electron transport chain during oxidative phosphorylation. It also metabolizes pyrimidines, fatty acids, and mitochondrial uncoupling proteins. In addition, coenzyme Q10 has been shown to mediate inflammation-related genes and participate in cholesterol metabolism (Hargreaves, Heaton & Mantle, 2020). Tsai et al. (2012) have demonstrated that CoQ10 ameliorates endothelial dysfunction by inhibiting the reduction in catalase activity induced by oxidized low-density lipoprotein (ox-LDL), thereby resulting in decreased ROS production in endothelial cells, increasing NO bioavailability, blocking cytochrome c release, and attenuating pro-apoptotic responses.

This network meta-analysis compared the effects of various nutritional supplements in PCOS patients. The results revealed that they could significantly improve various outcomes in PCOS patients, such as glucose and lipid metabolism. However, the intervention dosage and course of treatment across different studies for the same type of nutritional supplement were inconsistent, making it difficult to determine the optimal regimen. From a practical perspective, nutritional supplements can be used as adjuvant treatment, but their clinical effects are inconclusive. More clinical trials are warranted to further explore the therapeutic effects of these nutritional supplements. Therefore, nutritional supplements should be cautiously used in the management of glucose, lipid metabolism and endocrine status of PCOS. Future clinical studies should compare nutritional supplements with traditional drugs (e.g., letrozole, metformin) for treating PCOS, to understand the differences in the mechanisms and effects of these different drugs. Additionally, the selection of research objects also needs to be refined in future studies, for example, to explore the effect of nutritional supplements on PCOS patients of different ages (e.g., adolescence and childbearing age), or with endocrine diseases such as obesity, hyperlipidemia, and type 2 diabetes. Furthermore, the dosage and course of treatment of nutritional supplements in the treatment of PCOS require further exploration. The evaluation of safety and quality of life assessments should be emphasized in clinical trials as well.

This study has a few limitations: (1) No significant differences were observed in TT, SHBG, and CRP between nutritional supplements and placebo, probably due to small sample sizes, inconsistent intervention protocols, and different lengths of the treatment course. (2) The frequency of nutritional supplements used among the included studies was considerably different. For example, 12 studies used vitamin D, whereas only one study used carnitine and inositol, which may limit the interpretation of the results to some extent. (3) The dosage and regimen of the same nutritional supplement were not completely standardized across studies, which may have had an impact on the findings. (4) There was some publication bias in some of the outcome indicators, which may have affected the results. (5) In the safety analysis, 19 studies did not report adverse reactions, and almost all studies lacked follow-up after treatment, which may affect the judgment of the long-term safety and potential adverse effects of nutritional supplements.

Conclusions

In conclusion, among eight nutritional supplements including selenium, chromium, carnitine, inositol, coenzyme Q10, Omega-3, probiotics, and vitamin D, carnitine may help reduce body weight and BMI in patients with PCOS. Chromium supplementation may lower fasting insulin levels and insulin resistance in patients with PCOS. Omega-3 and selenium are advantageous in improving fasting blood glucose and QUICKI, respectively; and coenzyme Q10 has a better effect in improving lipid metabolism such as lowering the levels of TG, TC, and LDL-C. The results of this study may assist clinicians in selecting appropriate nutritional supplements for PCOS patients based on their individual differences and therapeutic goals. Meanwhile, more large-scale clinical randomized controlled trials are needed to validate the effects of nutritional supplements in PCOS patients.

Supplemental Information

Supplemental Information 1 PRISMA checklist.

Click here for additional data file.

Supplemental Information 2 Search strategy.

Click here for additional data file.

Supplemental Information 3 Systematic Review and Meta-Analysis Rationale.

Click here for additional data file.

Abbreviations

PCOS Polycystic ovary syndrome

BMI Body mass index

FBG Fasting blood glucose

FINF Fasting insulin

HOMA-IR Insulin resistance index

QUICKI Quantitative insulin sensitivity index

TG Triglycerides

TC Total cholesterol

LDL-C Low-density lipoprotein cholesterol

HDL-C High-density lipoprotein cholesterol

TT Total testosterone

SHBG Sex-hormone binding globulin

CRP C-reactive protein

TAC Total antioxidant capacity

MD Mean differences

MCMC Markov chain Monte Carlo

SUCRA Surface under the cumulative ranking probabilities

IRS1 Insulin receptor substrate 1

Glut 4 Glucose transporter 4

PI3-k Phosphatidylinositol 3-kinase

Akt Protein kinase B

JNK C-Jun N-terminal kinase

EPA Eicosapentaenoic acid

DHA Docosahexaenoic acid

TNF-α Tumor necrosis factor-α

IL-6 Interleukin-6

GLP-1 Glucagon-like peptide 1

ROS Reactive oxygen species

PCOS-IR Polycystic ovary syndrome-insulin resistance

SeNPs Selenium nanoparticles

PI3K/Akt Phosphatidylinositol 3-kinase/protein kinase B

ox-LDL Oxidized low-density lipoprotein

Additional Information and Declarations

Competing Interests

Author Contributions

Data Availability

The authors declare that they have no competing interests.

Xinyin Hu conceived and designed the experiments, performed the experiments, analyzed the data, prepared figures and/or tables, authored or reviewed drafts of the article, and approved the final draft.

Wanyi Wang conceived and designed the experiments, performed the experiments, analyzed the data, prepared figures and/or tables, authored or reviewed drafts of the article, and approved the final draft.

Xuhan Su performed the experiments, prepared figures and/or tables, and approved the final draft.

Haoye Peng performed the experiments, prepared figures and/or tables, and approved the final draft.

Zuolin Tan performed the experiments, authored or reviewed drafts of the article, and approved the final draft.

Yunqing Li performed the experiments, authored or reviewed drafts of the article, and approved the final draft.

Yuhua Huang conceived and designed the experiments, analyzed the data, authored or reviewed drafts of the article, and approved the final draft.

The following information was supplied regarding data availability:

This is a systematic review/meta-analysis.

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
