# Peer review of "Comparison of nutritional supplements in improving glycolipid metabolism and endocrine function in polycystic ovary syndrome: a systematic review and network meta-analysis"

_PeerJ, doi:10.7717/peerj.16410_

## Round 0.1 · original submission · Minor Revisions

Please respond and make appropriate revisions based on the reviewers' suggestions and my comments (below). This will greatly improve the quality of the manuscript.

Here are my comments:
1. Figures 2, 3, 4 and 5, are too small to be clearly distinguished by the readers. Please re-edit these figures to their final printed size.
2. Some important outcomes such as long-term safety, patient-reported outcomes, and potential adverse effects of the supplements are not fully addressed.
3. The study doesn't discuss the specific dosages, durations, and forms of nutritional supplements used in the interventions, which are important for understanding the practical implications.
4. In their future research, the authors should incorporate patient-reported outcomes and quality of life assessments to provide a more comprehensive understanding of the supplements' impact on PCOS patients.
5. [as some concerns] should be [as having some concerns]. Similarly, [as some concerns] should be also revised.

**Language Note:** The review process has identified that the English language must be improved. PeerJ can provide language editing services - please contact us at copyediting@peerj.com for pricing (be sure to provide your manuscript number and title). Alternatively, you should make your own arrangements to improve the language quality and provide details in your response letter. – PeerJ Staff

Reviewer 1 ·

Basic reporting

I appreciate the authors' effort in conducting a systematic review and network meta-analysis to evaluate the efficacy of nutritional supplements for treating polycystic ovary syndrome. This research topic holds significant importance within the field. However, my assessment of the manuscript reveals some critical issues that require immediate attention before considering it for publication.

Inconsistent Fonts and Figure Size: Figure 2 exhibits inconsistencies in fonts, and the figure itself is presented at an insufficient size, hindering its legibility. I recommend that the authors standardize fonts and ensure that all figures are sufficiently sized for clarity. Moreover, it is crucial to enhance the clarity of horizontal and vertical coordinates in figures 4 and 5 to facilitate a better understanding of the data presented.

Abbreviations in Abstract: It is essential that all abbreviations are spelled out upon their first appearance in the Abstract. This will improve the manuscript's accessibility and ensure that readers can fully grasp the content without prior knowledge of the terminology.

Language Quality: The standard of English throughout the manuscript is notably poor, making it challenging to comprehend certain sections. I strongly recommend that the authors engage a proficient English language editor or a professional language editing service. This will not only enhance the manuscript's readability but also elevate its scientific merit by ensuring that the content is expressed accurately and clearly.

In conclusion, while the research topic addressed in this manuscript is intriguing and carries substantial significance, these fundamental errors in presentation, abbreviation usage, and language quality significantly impact the overall quality of the paper. Addressing these concerns is crucial to elevate the manuscript's readiness for publication in a reputable scientific journal.

Experimental design

No obvious problem

Validity of the findings

I believe it has effectiveness and value

Additional comments

None

·

Basic reporting

I would like to provide a more comprehensive set of review comments to enhance the quality and depth of the feedback:

1. It is important to enhance the clarity and comprehensiveness of the materials and methods section by supplementing the keywords used in the study. Keywords play a crucial role in facilitating the discoverability and indexing of your research. Consider adding relevant keywords that align with the main research objectives and outcomes.

2. The discussion section should be structured in a way that begins with a concise presentation of the main results obtained from the study. This will help readers grasp the key findings immediately. Additionally, it would be beneficial to include a subsection specifically dedicated to the implications of the study for future research. Elaborate on how the findings contribute to the existing body of knowledge and suggest potential avenues for further investigation.

3. While the study provides valuable insights into the effectiveness of nutritional supplements for PCOS patients, it is essential to address the potential adverse reactions associated with these supplements. This information is crucial for a comprehensive understanding of the risks and benefits. Please provide details regarding any adverse reactions observed during the study and their associated implications for patient safety and management.

4. The manuscript would benefit significantly from a thorough proofreading and language editing. Numerous typos and grammatical errors throughout the manuscript detract from the overall readability and professionalism of the work. To enhance the manuscript's quality, I recommend a careful revision for clarity, grammar, and style. Consider enlisting the assistance of a professional language editor or proofreading service to address these issues effectively.

These comments, when expanded upon and integrated into the review, will provide more constructive and detailed feedback to the authors, ultimately assisting in the improvement of their research paper.

Experimental design

no comments

Validity of the findings

no comments

Additional comments

no comments

Reviewer 3 ·

Basic reporting

no comment

Experimental design

no comment

Validity of the findings

no comment

Additional comments

I would like to provide more comprehensive feedback on this paper, aiming to enhance the clarity and quality of the manuscript:

1) It is essential that all abbreviations used in the manuscript be introduced with their full names when first mentioned. For example, when referring to cytokines like IL-6 and TNF-α, please provide their complete names to ensure readers' comprehension and make the text more accessible.

2) In accordance with best practices in scientific reporting, it is crucial that the authors accurately present the 95% confidence intervals (95% CI) when discussing their findings. Properly citing the published literature to support their claims and using the appropriate format for expressing confidence intervals will bolster the paper's scientific rigor.

3) The overall quality of English language writing in the manuscript needs improvement. To enhance the readability and professional appearance of the paper, I recommend seeking assistance from a fluent English speaker or professional editing services. Ensuring that the language is clear and precise will contribute to the paper's overall effectiveness.

4) To maintain consistency and align with standard scientific writing conventions, all punctuation marks and symbols should be in English. This adjustment will enhance the paper's professionalism and readability.

5) It would be beneficial to explicitly state whether the outcome measures in the study are categorized as primary or secondary. This clarification will aid readers in understanding the hierarchy of outcomes and their relevance to the study's objectives.

6) In the discussion section, it is important for the authors to underscore the novelty of their findings. Elaborating on how their research contributes to existing knowledge and discussing the practical implications of their results in clinical settings will enrich the paper's value and significance.

7) To strengthen the paper's methodology and analytical rigor, I recommend that the authors consider adding sensitivity or specificity analyses, especially if these parameters are relevant to their research objectives. This additional analysis will enhance the comprehensiveness of the study.

---

## Round 0.2 · accepted · Accept

Because the reviewers and I shared most of the concerns, I independently reviewed the revised version, and have confirmed that all of the reviewers' comments and my concerns were appropriately addressed. I am satisfied with the latest version. I think this revised article can be considered for publication in PeerJ.